# Genome-Wide Identification of *GmSPS* Gene Family in Soybean and Expression Analysis in Response to Cold Stress

**DOI:** 10.3390/ijms241612878

**Published:** 2023-08-17

**Authors:** Jiafang Shen, Yiran Xu, Songli Yuan, Fuxiao Jin, Yi Huang, Haifeng Chen, Zhihui Shan, Zhonglu Yang, Shuilian Chen, Xinan Zhou, Chanjuan Zhang

**Affiliations:** 1Key Laboratory of Biology and Genetic Improvement of Oil Crops, Ministry of Agriculture and Rural Affairs, Oil Crops Research Institute of Chinese Academy of Agricultural Sciences, Wuhan 430062, China; 2College of Life Sciences, Wuhan University, Wuhan 430072, China

**Keywords:** soybean, sucrose phosphate synthase, genome-wide survey, gene expression, cold stress

## Abstract

Sucrose metabolism plays a critical role in development, stress response, and yield formation of plants. Sucrose phosphate synthase (SPS) is the key rate-limiting enzyme in the sucrose synthesis pathway. To date, genome-wide survey and comprehensive analysis of the *SPS* gene family in soybean (*Glycine max*) have yet to be performed. In this study, seven genes encoding SPS were identified in soybean genome. The structural characteristics, phylogenetics, tissue expression patterns, and cold stress response of these *GmSPSs* were investigated. A comparative phylogenetic analysis of SPS proteins in soybean, *Medicago truncatula*, *Medicago sativa*, *Lotus japonicus*, Arabidopsis, and rice revealed four families. GmSPSs were clustered into three families from A to C, and have undergone five segmental duplication events under purifying selection. All *GmSPS* genes had various expression patterns in different tissues, and family A members *GmSPS13*/*17* were highly expressed in nodules. Remarkably, all *GmSPS* promoters contain multiple low-temperature-responsive elements such as potential binding sites of inducer of CBF expression 1 (ICE1), the central regulator in cold response. qRT-PCR proved that these *GmSPS* genes, especially *GmSPS8*/*18*, were induced by cold treatment in soybean leaves, and the expression pattern of *GmICE1* under cold treatment was similar to that of *GmSPS8*/*18*. Further transient expression analysis in *Nicotiana benthamiana* and electrophoretic mobility shift assay (EMSA) indicated that *GmSPS8* and *GmSPS18* transcriptions were directly activated by GmICE1. Taken together, our findings may aid in future efforts to clarify the potential roles of *GmSPS* genes in response to cold stress in soybean.

## 1. Introduction

Environmental factors such as cold and osmotic stresses could affect plant growth. Cold stress includes chilling (0–15 °C) and freezing (<0 °C), which have an effect on the growth and yield of plants [1,2]. Low temperature affects the absorption of water and nutrients by plants, as well as the fluidity of membrane, and affects gene expression and protein synthesis [2]. It affects cell metabolism by reducing the rate of biochemical reaction or affecting gene expression reprogramming. Cold stress will not only inhibit the metabolic reaction of plants, but also produce osmotic reactions to inhibit plant growth [2,3]. In addition to winter-habit plants, many important crops such as rice, corn, and soybean are very sensitive to cold [4]. Cold stress will seriously harm the growth of soybean, especially spring soybean cultivars, and lead to yield reduction, so it is vital to reveal the adaptation mechanism of soybean to cold stress.

In order to survive, plants have evolved complex mechanisms in response to cold stress. In previous studies, many key factors in the cold stress signal pathway have been found. *COR* (cold-responsive) genes can be upregulated by *CBFs* (C-repeat/DRE-binding factors) under cold stress [4,5], which could be induced by low temperature and had conserved functions in flowering plants in response to cold stress [6]. Overexpression of *CBF1/DREB1b* and *CBF3* can enhance the freezing resistance of Arabidopsis [7,8]. ICE transcription factor can induce the expression of *CBF* genes under normal temperature, and the *ice1* mutant blocked the expression of *CBF3* itself and its downstream genes to increase the sensitivity of plants to cold stress in Arabidopsis [9]. These results indicate that the ICE-CBF-COR signal pathway is crucial for plants in response to cold stress.

Soybean is a worldwide economical crop and a major source of oil and protein [10]. Some studies have been reported cold stress responses in soybean. Under cold treatment, transcriptome analyses identified many cold-stress-related genes, including *CBF*/*DREB* [11]. In addition, the plant hormone ethylene may inhibit soybean CBF/DREB1 pathway through EIN3 (ethylene-insensitive 3) [12]. The sensitivity of soybean to cold stress varies at different developmental stages. During the vegetative growth stage, *CBF* can be strongly and briefly induced under cold stress, but low-temperature stress delays the reproductive stage of soybean [13,14]. *GmTCF1a* (tolerant to chilling and freezing 1a) positively regulates cold tolerance in soybean, which is independent of the CBF pathway as *AtTCF1* [15,16].

The physiological changes of plants are closely related to freezing resistance, including the accumulation of some solutes, such as soluble sugar, proline, and other lower-molecular-weight solutes [2]. The increase of soluble sugars such as sucrose, glucose, and fructose can function as osmotic protective substances to enhance the resistance of plants to cold [17]. A previous study showed that soluble sugar accumulated significantly within 2 h after plants were exposed to low temperature [18]. Prolonged cold will lead to more sucrose accumulation, which enables *Deschampsia antarctica* to survive in the Antarctic [19]. When petunia was cold-treated, more sugar and starch were accumulated in the source leaves, while the starch content in the sink tissue decreased. Moreover, the activity of cwINV (cell wall invertase) in the sink tissue decreased, showing the reduction of sugar input and utilization [20]. Accumulating higher sugar in the source tissue can increase osmotic protection and thus enhance the ability of photosynthetic tissue to resist cold stress, which is the core element of cold stress response [21,22]. These results suggest that sugar accumulation plays an important role in plant resistance to cold stress.

Sucrose is the main transport form of photosynthate in plants and plays a crucial role in the normal growth and development of plants [23]. Sucrose phosphate synthase plays a central role in the production of sucrose in photosynthetic cells. The sucrose synthesis involves a two-step reaction, and SPS (EC 2.4.1.14) is the key rate-limiting enzyme in the sucrose synthesis pathway, which catalyzes the conversion of UDP-glucose and fructose-6-phosphate to sucrose-6-phosphate, which is then hydrolyzed to sucrose by sucrose-phosphatase (SPP) [24,25]. SPS activity has been shown to be linked with sucrose accumulation [26]. Spinach *SPS* expression in cotton promotes sucrose synthesis and improves fiber quality [27]. At the stage of fruit ripening, both *SPS* expression and activity are upregulated, which in turn promotes sucrose synthesis [28,29,30]. SPS in spinach has also been proposed to play significant roles in stressful environment conditions. When plants are exposed to low-temperature stress, the expression of *SPS* increases dramatically and more sucrose is synthesized in response to cold stress [31]. SPS is encoded by a multigene family. Genome-wide survey of *SPS* genes has been performed in Arabidopsis, rice, maize, wheat, tomato, cassava, Litchi, and other plants [32,33,34,35,36]. According to the amino acid sequences, SPS proteins can be divided into four categories (A, B, C, D), with the branch D only present in the poaceae [37]. Expression and functions of different *SPS* genes vary among different categories in different plants. In soybean leaves, drought and K-deficiency increased soluble sugar content and SPS activity [38], and accumulation of sugar content was correlated with increased SPS activity, suggesting SPS might play important roles in response to abiotic stress in soybean [39].

In this study, a genome-wide survey was performed to explore the special sequence and expression characteristics of soybean *SPS* family genes. The objective focused on the analysis of phylogenetic relationships, duplications patterns, gene structures, conserved motifs, phosphorylation sites, cis-elements, and tissue expression patterns. The expression change of *GmSPS* genes under cold stress were also examined. Furthermore, we suggest that the expression level of *GmSPS8/18* may be upregulated by GmICE1. Consequently, these findings may provide foundations for functional investigation of *GmSPS* genes in soybean.

## 2. Results

### 2.1. Genome-Wide Identification of GmSPS Genes in Soybean

HMMER (v3.3.2) searches using three conserved domains (PF00862, PF00534, PF05116) of SPS protein and BLASTP using the Arabidopsis SPS sequences were performed in the soybean protein database. Then, 82 and 18 candidate sequences were generated through HMMER (v3.3.2) search and BLASTP, respectively. Finally, a total of seven different soybean loci encoding SPS proteins were identified by removing redundant sequences. The seven putative soybean SPS proteins all contained the sucrsPsyn_pln, and were confirmed by NCBI-CDD (Appendix A). Based on the annotation file of soybean genomic sequences, the seven putative soybean *SPS* genes were found to be distributed on seven different chromosomes, respectively. According to the chromosomal location, these soybean *SPS* genes were named *GmSPS4*/*6*/*8*/*13*/*14*/*17*/*18*.

Sequence alignments showed high identity and similarity in GmSPS4/6, GmSPS13/17, and GmSPS8/14/18 at both amino acid (79.29% to 97.26%) and nucleotide level (83.18% to 96.77%) (Appendix A). The detailed predictions of *GmSPS* genes are described in Table 1. The full length of GmSPS-predicted proteins vary from 778 to 1064 amino acids and the molecular weight (Mw) is arranged from 87.6 to 119.3 kDa. In addition, the predicted isoelectric point (pI) ranges from 5.94 to 6.31. More than one transcript were predicted in three *GmSPS* genes. *GmSPS13* contains five transcripts, *GmSPS14* contains two transcripts, and *GmSPS17* contains four transcripts. Moreover, subcellular localization prediction suggests that GmSPS4/8/14 and GmSPS6/13/17/18 proteins are located in the cytoplasm and nucleus, respectively.

### 2.2. Phylogenetic Analysis of SPS Proteins

A phylogenetic tree was constructed using the MEGA software (v11.0.11) to assess the phylogenetic relationship of SPSs from soybean with those from *M. truncatula*, *M. sativa*, *L. japonicus*, Arabidopsis, and rice, which are leguminous, dicotyledonous, and monocotyledonous model plants, respectively (Figure 1). Previous studies isolated three MsSPS members (MsSPSA, MsSPSB, MsSPSB3) in *M. sativa* [40]. However, we identified two MsSPSB3 members (named MsSPSB3-1 and MsSPSB3-2) in diploid *M. sativa,* which was sequenced in 2020 [41]. *MsSPSB3* amplified in previous studies was a partial of *MsSPSB3-1* and *MsSPSB3-2*. Thus, four SPSs (MsSPSA, MsSPSB, MsSPSB3-1, and MsSPSB3-2) from *M. sativa* were used to construct the phylogenetic tree. These SPS members from six plant species can be clustered into four distinct families (A, B, C, D). The family D is specific to rice, and the family C does not include alfalfa SPS members, which is consistent with previously studies [25,40]. Among the four families, family B had the largest number of SPS members, with twelve. The GmSPS had a closer relationship with leguminous plants than Arabidopsis and rice. In families A, B, and C, SPSs from soybean are more closely related to *L. japonicus* than two kinds of alfalfa. GmSPS13 and GmSPS17 were grouped into family A, GmSPS8, GmSPS14, and GmSPS18 belonged to family B, and GmSPS4 and GmSPS6 were grouped into family C. No SPS from soybean was grouped into family D. Both soybean and *L. japonicus* had SPS members of family C, but alfalfa did not.

### 2.3. Collinearity Analysis of GmSPS Genes

Soybean is a paleotetraploid crop which has undergone two whole genome duplications with very high retention of homologs in the genome [42,43]. Whole genome duplication and segmental and tandem duplication are the important events during soybean genome regions [42]. Seven *GmSPS* genes were not located in the same soybean chromosome. Therefore, no tandem duplication was found in the identified *GmSPS* genes. Collinearity analyses revealed that five gene pairs underwent segmental duplication events (Figure 2).

Genome duplications occurred at approximately 59 and 13 million years ago, resulting in a highly duplicated genome with nearly 75% of the genes present in multiple copies [42]. The Ks values of segment pairs *GmSPS13/17*, *GmSPS4/6,* and *GmSPS8/18* were 0.1107, 0.0995, and 0.1182 (Table 2). The divergence times of these duplicated paralogous pairs were associated with the 13 Mya WGD events. However, the Ks of *GmSPS18/14* and *GmSPS8/14* were larger than 0.6 and the divergence time is associated with the 59 Mya early legume WGD, which indicated that both *GmSPS8* and *GmSPS18* were originated from *GmSPS14* [44]. The Ka/Ks ratio is a measure used to explore the mechanism of gene replication and evolution after ancestor differentiation [45]. A Ka/Ks value < 1 suggests purifying selection, a Ka/Ks value = 1 indicates neutral selection, and a Ka/Ks value > 1 suggests positive selection. In addition, all the *GmSPS* paralogous pairs showed Ka/Ks < 1, suggesting that *GmSPS* genes have undergone purification selection during evolution.

### 2.4. Gene Structure and Conserved Motif Analyses of GmSPSs

The gene structures of seven *GmSPS* genes are shown by analyzing the sequence annotation file (Figure 3A). The number of exons and introns in *GmSPS* genes vary from twelve to seventeen and eleven to sixteen, respectively. *GmSPS* genes belonging to the same family possess similar number of exons and introns. Family A members, the putative paralogous pairs (*GmSPS13/17*), contain 13 exons and 12 introns, and family B members *GmSPS8/14/18* all contain 12 exons and 11 introns. Family C members, the putative paralogous pairs (*GmSPS4*/*6*), contain the most exons (17, 14) and introns (16, 13). Fifteen putative conserved motifs were identified in seven GmSPS proteins based on the amino acid sequence by the program MEME (Figure 3B). The identified multilevel consensus sequences of these motifs are shown in Appendix A. All GmSPS proteins contain these fifteen motifs except GmSPS4, which not contain motif 6, motif 8, and motif 9. Most motifs are present in Glycos_transf_1, S6PP and Sucrose_synth domains (Appendix A). Motif 5, motif 15, motif 9, motif 4, motif 12, and motif 3 belonged to the sucrose synthase domain. Motif 2, motif 6, motif 10, and motif 1 belonged to the Glycos_transf_1 domain. Motif 13, motif 11, motif 14, and motif 7 belonged to the S6PP domain. Glycos_transf_1 and S6PP domains existed in all GmSPS members. It is worth noting that GmSPS13 did not have the Sucrose_synth domain by using Pfam for functional domain prediction, although it had conserved motifs. This may mean that the Sucrose_synth domain function was missing in GmSPS13. The Glycos_transf_1 domain is associated with transfer of glucosyl, and the S6PP domain may mediate the interaction with SPP.

### 2.5. Analysis of Phosphorylation Sites in GmSPS Proteins

It has been reported that SPS proteins could be modulated by phosphorylation in response to temperature and other abiotic stresses [37]. The predicted phosphorylation sites of GmSPS proteins were analyzed using NetPhos 3.1 Server (Figure 4 and Appendix A). The results show that the main predicted phosphorylation site of GmSPS proteins was serine. In detail, GmSPS13 had the most predicted serine phosphorylation sites with a total number of 90, and GmSPS4 had the least number of 55. GmSPS14 had the maximum number of predicted threonine phosphorylation sites and the GmSPS6 contained the maximum number of predicted tyrosine phosphorylation sites. Previous research indicated that phosphorylation sites Ser-158, Ser-229, and Ser-424 were involved in light–dark regulation, 14-3-3 protein binding, and osmotic stress activation, respectively [37]. Our results demonstrate that Ser-158 is conserved in GmSPS proteins except GmSPS4. Ser-229 is conserved in GmSPS proteins apart from family C members GmSPS4 and GmSPS6. A conversion from S to A was found in GmSPS4 and GmSPS6, which is consistent with the fact that in LcSPS4, the family C member from Litchi is observed [36]. Ser-424 is conserved in GmSPS proteins except family B members GmSPS8, GmSPS14, and GmSPS18 (Figure 4B). A conversion from S to N was found in GmSPS8, GmSPS14, and GmSPS18, which is consistent with the fact that in AtSPS3 and LcSPS3, the family B member from Arabidopsis and Litchi [36], respectively, are observed. Accordingly, our data suggest that GmSPSs contain several predicted phosphorylation sites including a part of serine, threonine, and tyrosine sites, which could possibly function in response to the abiotic stresses.

### 2.6. Promoter cis-Element Analysis of GmSPS

Gene promoters are essential for transcriptional regulation [46]. The cis-elements of promoters play significant roles in response to plant growth and various environmental stresses. The 3000 bp sequences upstream of all *GmSPS* genes were analyzed by PlantCARE and PlantPAN (Appendix A). Some kinds of cis-elements were found in the promoter regions of *GmSPSs,* including TF binding sites, hormone response, abiotic response, and photoperiod response elements (Figure 5). Predicted cis-elements that could respond to the hormones such as auxin, gibberellin, salicylic acid, abscisic acid, and MeJA (methyl jasmonate) were identified. Some elements associated with abiotic stresses response like defense, low temperature, drought-inducibility, and anaerobic induction in upstream regions of the *GmSPS* genes were also predicted. It is worth noting that all the *GmSPS* members have many low-temperature-responsive elements, and they are closer to the predicted start codon of family B members (*GmSPS8*/*14*/*18*) than other *GmSPSs*. These results suggest that *GmSPS* genes may play important roles in response to environmental factors, especially low-temperature stress.

### 2.7. Tissue-Specific Expression Patterns of GmSPS Members in Soybean

To further clarify the expression pattern of *GmSPSs* in soybean, the expression levels of *GmSPSs* in eight different tissues including root, stem, leaf, petiole, flower, SAM (shoot apical meristem), pod, and mature seed were analyzed by qRT-PCR. The relative mRNA abundance of *GmSPS* genes was shown in a heatmap (Figure 6A). The results demonstrated that all *GmSPS* genes were expressed in stem, leaf, petiole, flower, and SAM, while not expressed in root and seed, except *GmSPS13* and *GmSPS17*. Our analysis revealed that *GmSPS8* had the strongest expression level in the flower. In contrast, *GmSPS6/13* and *GmSPS17* had higher expression in pod and petiole compared to other tissues, respectively. The expression patterns may lay the foundation for exploring *GmSPSs* potential functions in the future.

In our previous work, we carried out the transcriptome analysis of soybean nodules at soybean different five developmental stages (branching stage, flowering stage, fruiting stage, pod stage, and harvest stage) inoculated with *Bradyrhizobium japonicum* strain 113-2 [47]. We analyzed the expression abundance of *GmSPS* genes using the published data (Figure 6B) and found significant differences in the expression level of *GmSPS* genes in soybean nodules. Interestingly, the family A members *GmSPS13/17* were highly expressed in nodules at all five stages. This is consistent with previous research showing that *MsSPSA* revealed nodule-enhanced expression [40]. The family B members, *GmSPS8/18,* were not expressed in nodules at any stage, while *GmSPS14* had some expression level in nodules at flowering stage to harvest stage. In addition, the family C member *GmSPS4* was not expressed in nodules, while the paralogous gene *GmSPS6* had some expression level in nodules at five stages. These results suggest function divergence of some homologous genes.

### 2.8. Expression Analysis of GmSPS Genes in Response to Cold Stress

It was predicted that a variety of low-temperature-responsive elements would be found in the *GmSPS* promoters (Figure 5). Moreover, previous research has suggested that SPS plays a key role in abiotic stresses, including cold [31,48,49,50]. To verify whether *GmSPS* genes responded to cold stress, the expression levels of *GmSPSs* were examined in leaves after cold treatment at 0 h, 2 h, 4 h, 6 h, 8 h, 12 h, 24 h, and 48 h. All *GmSPSs* were found to be upregulated under cold treatment (Figure 7). The expression levels of *GmSPS4/6* increased significantly at 8 h and reached the highest value at 12 h after cold treatment. The transcript levels of *GmSPS8/18* began to accumulate after 2 h after cold treatment, and increased 8-fold at 4 h and more than 20-fold from 6 h to 12 h. The levels of *GmSPS13/17* transcripts increased at 6 h and gradually declined from 8 h after cold treatment. The transcript level of *GmSPS14* increased over 10-fold at 6 h after cold treatment. Consequently, our results reveal the potential vital biological functions of *GmSPSs* for cold response.

### 2.9. GmSPS8/18 Were Upregulated by GmICE1 Involved in Cold Stress

The expression of *GmSPS* genes, especially *GmSPS8* and *GmSPS18*, were significantly upregulated after cold treatment. It was noted that multiple potential binding sites of transcription factor ICE1 existed in the promoter region of *GmSPS* genes. ICE1 is a central regulator in response to cold stress in plants [9], and GmICE1 is the homolog in soybean. We examined the transcript level of *GmICE1* to check whether *GmICE1* and *GmSPS* genes are in a coexpression network. We found that the transcript abundance of *GmICE1* was upregulated and reached a peak at 12 h under cold treatment (Figure 8A), which is similar to that of *GmSPS8* and *GmSPS18*. Whether GmICE1 could directly regulate the expression of *GmSPS8* and *GmSPS18* genes was investigated in transiently transformed tobacco leaves using a GUS reporter assay. The results indicated that tobacco leaves cotransformed with GmICE1 and any one of two soybean promoters produced three- to seven-fold higher value of GUS activity than the control tobacco leaves only transformed with the soybean promoters without GmICE1 (Figure 8B). To verify whether GmICE1 could bind to the promoters of *GmSPS8* and *GmSPS18* in vitro, EMSAs were conducted. The biotin labeled 50 bp DNA fragment containing CACGTG elements (the putative binding site of ICE1) in *GmSPS8* promoter (−66 bp to −115 bp) and in *GmSPS18* promoter (−68 bp to −117 bp) were used as probes. Results revealed that recombinant GmICE1-His could bind the biotin-labeled probes, which was weakened by the unlabeled probes in a dose-dependent manner. These data demonstrated that GmICE1 directly bound to *GmSPS8* and *GmSPS18* promoters (Figure 8C). Taken together, these results confirm that GmICE1 could bind to the promoter regions of *GmSPS8* and *GmSPS18* in vitro and might activate their transcriptions under cold stress.

## 3. Discussion

Extreme weather such as low temperature can have a serious impact on the growth of soybean. Previous research showed that the sensitivity of soybean to cold stress varies at different developmental stages [13,14]. Many cold-stress-related genes have been identified through transcriptome analysis in soybean, including *CBF/DREB* [13]. The ethylene signaling pathway may negatively impact soybean CBF/DREB-regulated cold response by EIN3 [12]. *GmTCF1a* regulates cold tolerance in soybean and is independent of the CBF pathway [15], but the mechanism underlying cold stress response in soybean remains unclear. In general, sucrose level is increased during cold response of plants. SPS is the key rate-limiting enzyme in the sucrose synthesis pathway in plants [27,28]. Here, we conducted the genome-wide survey of soybean SPS family genes and analyzed their response to cold stress, and our findings showed that the expression of all *GmSPS* genes increased significantly under cold stress, especially for *GmSPS8/18,* which can be directly activated by *GmICE1* in *N. benthamiana*. These results provide research ideas and clues for studying soybean cold stress response.

We assessed the phylogenetic relationships of SPS proteins between soybean and other plant species. As previously reported [33,37], all SPS proteins were divided into four families; only SPS proteins from rice belonged to family D. In families A, B, and C, the SPS proteins of legume species were clustered into a small branch, and the SPS proteins of soybean had a closer relationship with that of lotus plants than alfalfa. Different to soybean and lotus, no alfalfa SPS proteins belonged to family C, which was consistent with a previous report [40]. As an ancient tetraploid, soybean has undergone two whole genome duplication events, including an ancient duplication prior to the divergence of papilionoid (58 Mya to 60 Mya) and a Glycine-specific duplication (13 Mya), which resulted in about 75% of the genes being paralogous genes [43]. Three paralogous pairs (*GmSPS13*/*17*, *GmSPS4*/*6*, and *GmSPS8*/*18*) were identified in soybean, and their divergence time was associated with the 13 Mya WGD events. Our analysis shows that *GmSPS* genes have undergone purification selection during evolution.

The tissue-specific expression pattern was analyzed to understand the potential functions of *GmSPS* genes. All *GmSPS* genes are expressed in stem, leaf, petiole, flower, and SAM, but little in root and seed. *GmSPS8* has the highest transcript abundance in flower, indicating that itmay play an important function in flower development in soybean. Moreover, the expression level of *GmSPS* genes in nodules at different developmental stages of soybean were analyzed. The interesting finding was that members of family A (*GmSPS13*/*17*) were significantly expressed at branching stage, flowering stage, and fruiting stage, the critical period of symbiotic nitrogen fixation. As the ortholog gene of *GmSPS13*/*17* in alfalfa, *SPSA* also showed nodule-enhanced expression [40] and was involved in the synthesis of sucrose in nodules [51]. These results indicated that legume SPS members of family A may have the conserved function in carbon metabolism in nodules.

According to the cis-acting elements analysis, *GmSPS* genes may be involved in response to cold stress. The transcript abundance of all seven *GmSPS* genes was upregulated in response to cold stress. The expression level of soybean *SPS* genes in family B (*GmSPS8*/*14*/*18*) and family C (*GmSPS4*/*6*) was significantly increased by more than 10 times, and *GmSPS18* had the greatest increase (53 times). The transcript level of *GmSPS17* was just increased by 2.6 times, and *GmSPS13* was slightly upregulated. Our results are similar to the previous studies, which showed that low temperature increased the content and expression level of *SPS* in chilling-sensitive maize and alfalfa, respectively [52,53]. The transcription levels of many genes are affected under low-temperature stress [54]. Several studies have demonstrated that the ICE1-CBF pathway performs a key role during cold acclimation [55]. In previous studies, the ChIP-seq (chromatin immunoprecipitation sequencing) data of ICE1 indicated that it can bind to the promoter regions of many cold-responsive genes, including *CBF* and a large number of *COR* genes [56]. However, it is unknown whether ICE1 could regulate the expression level of *SPS* genes. Our data suggest that the expression levels of *GmSPS8* and *GmSPS18* could be activated by GmICE1 using transient expression GUS reporter assays in *N. benthamiana*. Additionally, the result of EMSA indicates that GmICE1 can directly bind to the promoter regions of *GmSPS8/18* in vitro. The PlantPAN prediction results showed that there was no CBF binding site in the promoters of *GmSPS8/18*. In addition, in Arabidopsis, the expression of *AtSPS3* (the orthologous gene of *GmSPS8/18*) did not significantly change in the *cbf123* triple mutant [57]. It is possible that *GmSPS8/18* participates in cold stress regulation independently of the CBF pathway in soybean. This may be a new mechanism for soybean plants to resist cold stress.

## 4. Materials and Methods

### 4.1. Identification of GmSPS Genes in Soybean

The four known Arabidopsis SPS (AtSPS1/2/3/4) amino acid sequences were extracted from the protein sequence file downloaded from the TAIR site (https://www.arabidopsis.org/) (accessed on 13 June 2022) and then were aligned to a soybean protein sequence file downloaded from the Soybase site (https://www.soybase.org/) (accessed on 13 June 2022) using BLASTP with default parameters [58,59]. In the meantime, the hidden Markov model (HMM) profile files of the SPS conserved domains Sucrose_synth (PF00862), Glycos_transf_1 (PF00534), and S6PP (PF05116) were downloaded from the Pfam site (http://pfam.xfam.org/) (accessed on 13 June 2022) [60]. HMMER (v3.3.2) was used to search the three conserved domains containing proteins in the soybean protein database with default parameter. Subsequently, the candidate soybean SPS proteins which contain the sucrsPsyn_pln were validated with NCBI-CDD (https://www.ncbi.nlm.nih.gov/cdd/) (accessed on 22 July 2022) (Appendix A).

### 4.2. Phylogenetic Analysis of SPS Proteins

To perform the phylogenetic analysis, we integrated SPS protein sequences of soybean, *M. truncatula*, *M. sativa*, *L. japonicus*, Arabidopsis, and rice. The SPS protein sequences of *M. truncatula*, *L. japonicus,* and rice were extracted from Phytozome v12 (https://phytozome.jgi.doe.gov) (accessed on 26 August 2022) [25,61]. The *M. sativa* SPS protein sequences were obtained from https://figshare.com/articles/dataset/Medicago_sativa_genomic_fa_zip/12859889 (accessed on 26 August 2022) [41]. The accession numbers of these *SPSs* are presented in Appendix A. PRANK software (v170427) was used to generate multiple alignments of SPS protein sequences. Next, the alignment results were imported into MEGA11 [62] for phylogenetic analysis using neighbor joining method with JTT + G model and 1000 bootstrap replicates.

### 4.3. Analysis of Gene Structure and Conserved Motifs

The gene features of *GmSPSs* were obtained from the annotation file of soybean. In addition, the conserved motifs of seven soybean SPS protein sequences were analyzed by MEME in the MEME-Suite site (https://meme-suite.org/meme/index.html) (accessed on 16 August 2022) [63]. TBtools (v1.120) was used to show the results of gene structure and conserved motifs [64].

### 4.4. Phosphorylation Sites Analysis in GmSPSs

Seven soybean SPS protein sequences were uploaded to Netphos 3.1 Server (https://services.healthtech.dtu.dk/service.php?NetPhos-3.1) (accessed on 15 August 2022) for phosphorylation sites analysis [65]. The numbers of identified serine, threonine, and tyrosine phosphorylation sites were calculated by the in-house Python script. TBtools was used to draw the heatmaps of phosphorylation sites.

### 4.5. Chromosomal Location and Collinearity Analysis

The chromosomal location information of all *GmSPS* genes was obtained from the annotation file of soybean. Furthermore, the chromosomal location figure of *GmSPS* genes was generated by TBtools. The longest protein-coding transcripts of *SPS* genes in soybean were screened, and the corresponding protein sequences were extracted. The genome-wide collinearity analysis was carried out by McscanX [66]. Tbtools was used to show the collinearity results. WGD segments pairs were verified using duplicate gene classifier in MCScanX. CDS alignments of each duplicate gene pair of GmSPSs were generated by ParaAT v2 with parameter “-m mafft -f axt -t -g”. Next, KaKs_Calculator 3.0 was used to calculate Ka (nonsynonymous) and Ks (synonymous) using CDS alignment file [67,68]. The divergence time of duplication pairs was calculated using T = Ks/(2 × 6.1 × 10^−9^) × 10^−6^ Mya [69].

### 4.6. cis-Regulatory Element in the Promoter Regions

The cis-regulatory element analyses of 3000 bp upstream sequences from the translation start codon of all *GmSPS* genes were generated by both PlantCARE (http://bioinformatics.psb.ugent.be/webtools/plantcare/html/) (accessed on 6 November 2022) [70] and PlantPAN (http://plantpan.itps.ncku.edu.tw/promoter_multiple.php) (accessed on 6 November 2022) sites [71]. The putative cis-regulatory elements related to resistance, hormones regulatory, transcription factor binding, etc., were displayed using TBtools.

### 4.7. Plant Materials and Cold Stress Treatment

At five days after germination, the seedlings of soybean cultivar Tianlong No.1 (TL-1, which was bred by our lab) were transformed to Hoagland nutrient solution under 60% relative humidity with a 12 h/12 h (light/dark) photoperiod at 25 °C. For cold treatment, seedings (V1 stage) were transferred to a low-temperature incubator at 4 °C. The first fully expanded leaves were collected at 0 h, 2 h, 4 h, 6 h, 8 h, 12 h, 24 h, and 48 h after treatment, and immediately frozen in liquid nitrogen then stored at −80 °C for RNA extraction. At each time point of treatment, six plants were harvested, with three biological replicates per sample. Untreated seedlings were used as control.

### 4.8. RNA Extraction and qRT-PCR Analysis

The qRT-PCR assay was used to analyze the expression pattern of *GmSPS* genes in various tissues of soybean cultivar TL-1 and soybean leaves under cold treatment. Total RNA was extracted using a TRIpure reagent (Aidlab, Beijing, China) according to the manufacturer’s instruction. The quantity of the extracted RNA samples was further assessed by agarose gel and nanospectrophotometer. The Hiscript II 1st strand cDNA synthesis kit (Vazyme, Nanjing, China) was used to synthesize the first strand of cDNA. The qRT-PCR was performed using iTaq Universal SYBR Green Supermix (Bio-Rad, Hercules, CA, USA) on a Bio-Rad CFX96 Real-Time PCR system. The gene-specific primers of *GmSPSs* were designed using the primer designing tool of NCBI (https://www.ncbi.nlm.nih.gov/tools/primer-blast/index.cgi) (accessed on 15 June 2022). Meanwhile, *GmActin11* (*Glyma.18G290800*), the expression of which did not change significantly in response to cold, was used as the internal reference gene to adjust the expression level of *GmSPS* genes. The primers are listed in Appendix A. All primers were synthesized by tsingke Biotech (Beijing, China). The qRT-PCR reactions were performed in a 20 μL volume and the cycling program conditions were denaturation at 95 °C for 5 min, followed by 40 cycles of 95 °C for 10 s, 60 °C for 10 s, and 72 °C for 20 s. All samples were performed with three technical and biological repeats. The 2^−ΔΔCt^ method [72] was utilized to analyze the relative expression level of *GmSPS* genes under cold stress.

### 4.9. GUS Activity Assay in Transiently Transformed N. benthamiana Leaves

The promoter sequences of *GmSPS8* (1956 bp upstream of the ATG) and *GmSPS18* (1950 bp upstream of the ATG) were cloned into the *Pst*I/*Bgl*II and *Bam*HI/*Nco*I restriction sites upstream of GUS reporter in the pCAMBIA3301 vector as the reporter constructs. The *GmICE1* CDS was constructed into PTF101 vector downstream of the *35S* promoter as the effector through homologous recombination. The constructed vectors were transformed into *Agrobacterium tumefaciens* EHA105 (Tsingke, Beijing, China). Then, the reporters and the effector were transient-expressed in *N. benthamiana* leaves as previously described [73]. These experiments were repeated at least three times independently. Primers are listed in Appendix A.

### 4.10. Electrophoretic Mobility Shift Assay

The EMSA was performed by generally following the previous method described [74]. The conserved domain sequences (826 bp to 1398 bp) of *GmICE1* were cloned into the PET-32a vector. The constructed vector was transformed into the *E. coli* BL21 (DE3) strain. The recombined protein of GmICE1 was purified using Ni-NTA Agarose (Qiagen, Dusseldorf, Germany). The DNA probes of *GmSPS8* and *GmSPS18* with biotin label were synthesized by Tsingke. The EMSA probes were obtained through annealing and renaturation of forward and reverse primers. EMSAs were performed using an EMSA Kit (Thermo Fisher Scientific, Waltham, MA, USA). Sequences of the probes and primers are listed in Appendix A.

## 5. Conclusions

In this study, we identified seven *SPS* genes in soybean genome via genome-wide screening. We carried out a series of analyses, including the phylogenetic relationships, duplications patterns, gene structures, conserved motifs, phosphorylation site, cis-elements, tissue expression patterns, and the expression of *GmSPS* genes in response to cold stress. We further demonstrated that *GmSPS8*/*18* can be directly activated by *GmICE1*. These data indicate that *GmSPS8/18* play important roles under cold stress. These investigations and analyses could increase our knowledge of the functions of SPS family genes in response to cold stress.

## Figures and Tables

**Figure 1 ijms-24-12878-f001:**
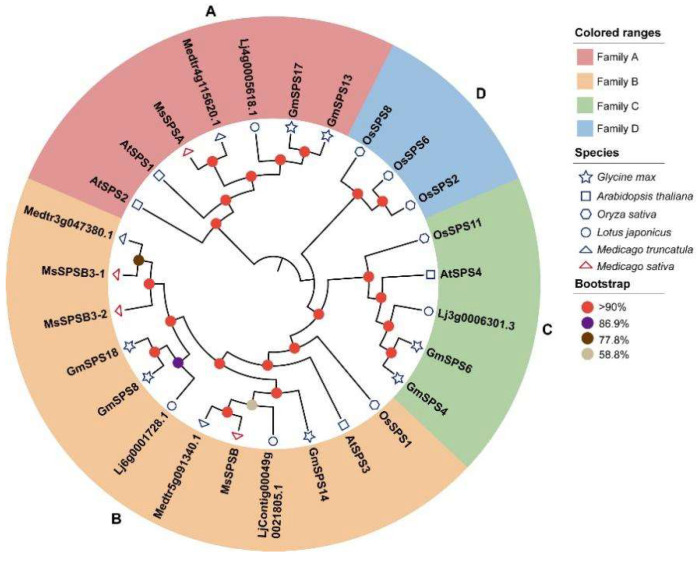
Phylogenetic analysis of SPS proteins. The tree was constructed and relied on the amino acid sequences by MEGA11 using neighbor joining method with JTT + G model and 1000 bootstrap replicates. Solid small circles with different colors are standard for bootstrap values. The tree can be categorized into four groups with different colors. The red, orange, green, and blue colors represent A–D groups, respectively. Different shapes indicate different species.

**Figure 2 ijms-24-12878-f002:**
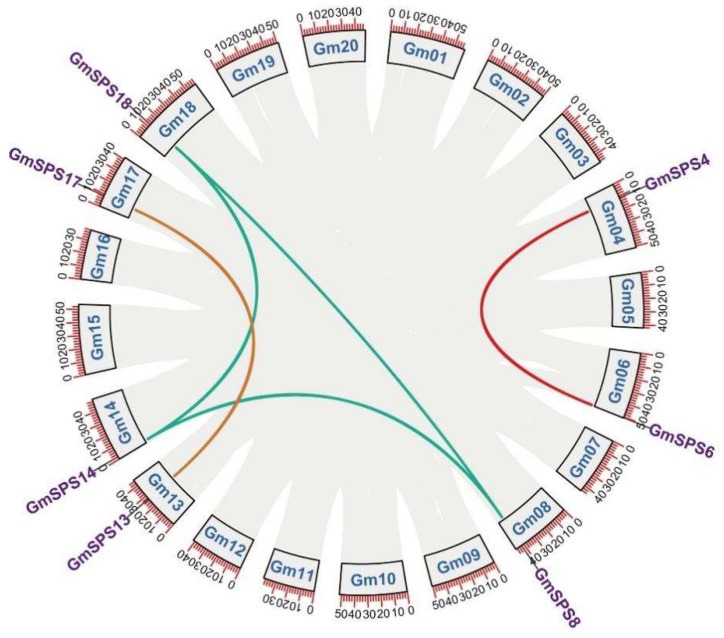
Collinearity analysis of *SPS* genes in soybean. The different colored lines connect two genes with collinearity. The gray lines represent other collinearity regions in the genome of soybean.

**Figure 3 ijms-24-12878-f003:**
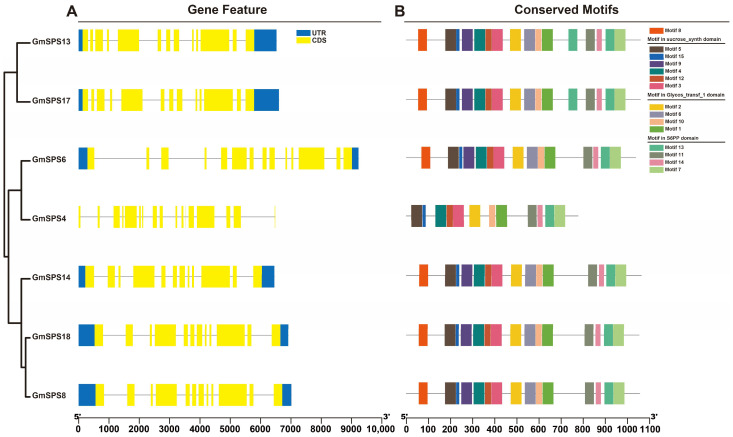
The gene structure and conserved motifs of *GmSPSs*. (**A**) The gene structure of *GmSPS* genes. The *GmSPSs* classified into three groups based on the phylogenetic relationships. The filled boxes and lines represent exons and introns, respectively. The blue and yellow boxes represent UTR and CDS, respectively. (**B**) Conserved motifs in GmSPS proteins. Boxes with different colors represent 15 different conserved motifs. The position and size of *GmSPS* genes and the corresponding proteins can be estimated by the scale at the bottom.

**Figure 4 ijms-24-12878-f004:**
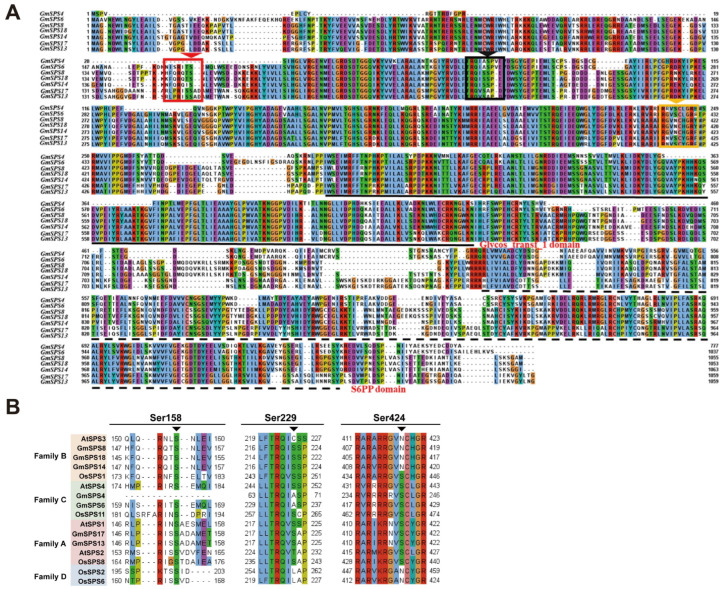
Phosphorylation sites analyses in SPS proteins. (**A**) The protein sequences alignment of GmSPSs. The red, black, and yellow arrows indicate Ser-158, Ser-229, and Ser-424 phosphorylation sites, respectively. Glycos transf_1 domain and S6PP domain are marked with black solid line and black dotted line, respectively. The alignment sequences are shown in jalview using Clustal colourscheme. (**B**) Ser-158, Ser-229, and Ser-424 phosphorylation sites in SPS families of soybean, Arabidopsis, and rice.

**Figure 5 ijms-24-12878-f005:**
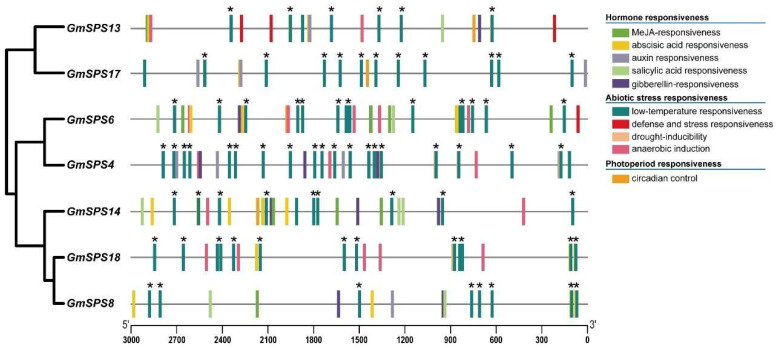
The cis-element analysis of *GmSPSs* promoter regions. Different colored boxes represent various elements. These asterisks represent potential binding sites of transcription factor ICE1. The positions of the cis-elements are shown at the bottom.

**Figure 6 ijms-24-12878-f006:**
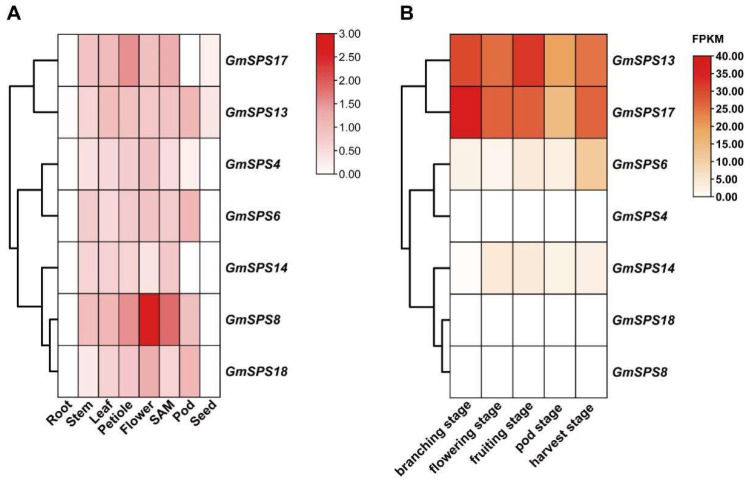
Expression profile of seven *GmSPS* genes in different tissues of soybean. (**A**) Expression patterns of *GmSPS* genes in soybean tissues. The expression value was obtained by qRT-PCR. At the right of the figure, different colors indicate the gene transcript abundance values. (**B**) Expression patterns of *GmSPS* genes in soybean nodules at different stages. FPKM values of *GmSPS* genes were obtained from published data [47]. Different colors of the heat map indicate the gene transcript abundance values.

**Figure 7 ijms-24-12878-f007:**
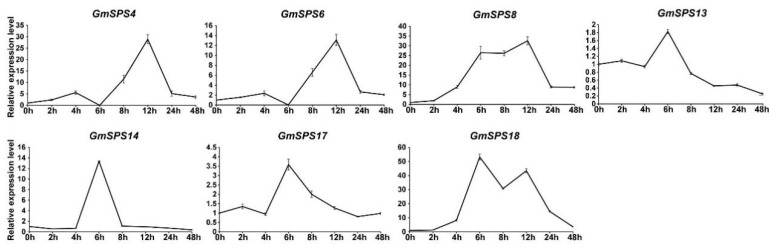
Relative expression levels of *GmSPS* genes in response to cold stress. RNA was extracted from treated leaves at 0 h, 2 h, 4 h, 6 h, 8 h, 12 h, 24 h, and 48 h, and qRT-PCR was performed using the specific primers of *GmSPS* genes. *GmActin11* was used as the internal reference. Treatments at each time point have their own controls, and expression at 0 h was set as “1”. The results represent the mean ± SD of three independent biological repetitions.

**Figure 8 ijms-24-12878-f008:**
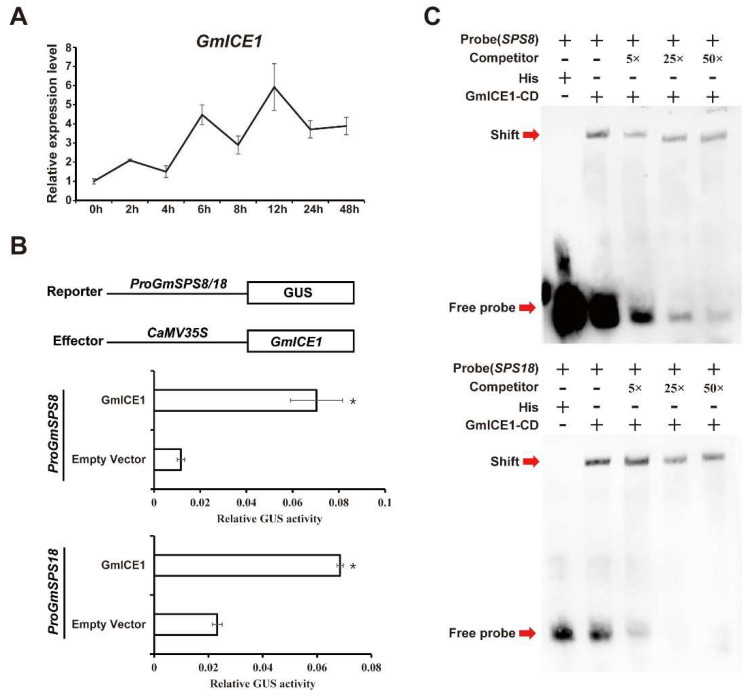
*GmSPS8* and *GmSPS18* transcriptions were directly activated by GmICE1. (**A**) Relative expression level of *GmICE1* in response to cold stress. The expression value was obtained by qRT-PCR. *GmActin11* was used as the internal reference. (**B**) GmICE1 promotes transcription of *GmSPS8/18*. The promoters of *GmSPS8* and *GmSPS18* are inserted in GUS reporter vector; in the meantime, the effector vector contains GmICE1. The vectors were coinfiltrated into *N. benthamiana* leaves to analyze the GUS activity. These results represent the mean ± SD of three independent biological repetitions. Asterisks represent the significant difference as determined by Student’s *t*-test (* *p* < 0.05). (**C**) EMSA showed the binding of GmICE1 to the promoters of *GmSPS8* and *GmSPS18* in vitro. The conserved domain of GmICE1 was expressed in *Escherichia coli* BL21 (DE3) cells to produce His-tagged GmICE1 protein. The recombinant protein was purified by Ni-NTA Agarose. Probe indicates DNA sequence with biotin label and the competitor is the same as the DNA sequence but without the biotin label. Red arrows indicate the positions of protein-probe complexes or free probes.

**Table 1 ijms-24-12878-t001:** All seven *GmSPS* genes, including their genome location and physical properties.

Gene Name	Gene ID	Genome Location	Transcript Numbers	Protein Size (aa)	MW (kDa)	pI	Subcellular Location
*GmSPS4*	Glyma.04g110200	Chr04: 11369193-11375676	1	778	87.58	5.99	Cytoplasm
*GmSPS6*	Glyma.06g323700	Gm06: 50675372-50684602	1	1038	117.09	6.31	Nucleus
*GmSPS8*	Glyma.08g308600	Gm08: 42128704-42135717	1	1056	118.80	6.18	Cytoplasm
*GmSPS13*	Glyma.13g161600	Gm13: 27135652-27142180	5	1060	118.01	6.04	Nucleus
*GmSPS14*	Glyma.14g029100	Gm14: 2121804-2128414	2	1064	119.33	5.94	Cytoplasm
*GmSPS17*	Glyma.17g109700	Gm17: 8599983-8606617	4	1060	118.08	6.09	Nucleus
*GmSPS18*	Glyma.18g108100	Gm18: 12235657-12242570	1	1054	118.39	6.1	Nucleus

**Table 2 ijms-24-12878-t002:** Identification of substitution rates for homologous *GmSPS* genes.

Segmental Duplicated Genes	Method	Ka	Ks	Ka/Ks	T (Mya)
*GmSPS17-GmSPS13*	MA	0.012484	0.110769	0.112704	9.079426
*GmSPS18-GmSPS14*	MA	0.074892	0.634249	0.118079	51.98762
*GmSPS6-GmSPS4*	MA	0.060117	0.099571	0.60376	8.161582
*GmSPS8-GmSPS14*	MA	0.068167	0.640031	0.106505	52.46156
*GmSPS8-GmSPS18*	MA	0.014738	0.118258	0.124628	9.693279

Ka: non-syn substitution rate, Ks: syn substitution rate, T: duplication date.

## Data Availability

The datasets used and/or analyzed in this study are available on reasonable request from the corresponding author.

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
