# Peer review of "Genome-Wide Identification of GmSPS Gene Family in Soybean and Expression Analysis in Response to Cold Stress"

_ijms, 2023, doi:10.3390/ijms241612878_

Round 1

Reviewer 1 Report

The authors have performed a very good depp analysis of the expression and behaviour of the sucrose phosphate synthase (SPS) in soybean. These enzymes have a key role in the sucrose synthesis pathway and, as explaines by the authors, they might have a role in cold acclimation responses in plants. Soybean is a complicated model because the two main genome duplication events, that usually result in duplicated genes and wide gene families. However, the authors have addressed this problem perfectly and they were able to analyze the SPS gene family adecuately. In my opnion the gene expression analysis, as performed, was very nice and easy to follow. 

My only problem is regarding the ICE1 role in the cold-specific activation of GmSPS8 and SPS18 genes. Because of the pressence of a putative ICE1 binding sequence, the authors explored via transient expression analysis the effect of the Co-expression of GmICE1 with their genes and monitoring GUS activity. In principle, the idea is right. however, to reach the conclusion that ICE1 is a transcriptional factor involved in the differential control of these two SPS genes other experiments should be neccessary.  In the absence of a ice1 ko mutant background in soybean, in which to analyze ICE1 and SPS8 or SPS18 expression, authors should have performed EMSA assays with a recombinant ICE1 protein to test whether ICE1 protein binds in vitro to these two promoters specifically. Other SPS gene promoters, not activated by ICE1 should be used as negative controls. In Arabidopsis, ChIP experiments could also be performed. I understand that in soybean they are pretty much complicated because of the absence of a specific GmICE1 protein antibody or the availability of GmICE1-GFP transgenic lines. But an appropiate EMSA assay is possible and should be performed by the authors.

Reviewer 2 Report

This manuscript is generally well-prepared and experimentally conducted, provides new information about SPS genes, expression, and predicted protein structure; and their cold regulation. 

However, the authors should better address previous work published on cold impact on soybeans.  A short para summarizing what is known about cold stress responses in soybean would make sense in the introduction. A better literature search should be done. 

For example: Other papers addressing expression under cold conditions.

1)    Maruyama et al, Plant Physiology, August 2009, Vol. 150, pp. 1972–1980 Also much is discussed that is relevant to sucrose metabolism related to cold (and drought) in Arabidopsis in this paper. (also see microarray data mentioned latter) 

2)    Robison et al., Front. Plant Sci., 12 February 2019 ABA and cold responses in soybean 

3)    Yamasaki et al 2016, cold responses in soybean (also see RNASEQ data mentioned alter)

4)    Wang et al., Growth and photosynthetic responses of soybean to short-term cold temperature Environmental and Experimental Botany

Volume 37, Issue 1, May 1997, Pages13-24

5)    Dong et al., BMC Plant Biol 2021 Aug 12;21(1):369 Enhancement of plant cold tolerance by soybean RCC1 family gene GmTCF1a

It would also make sense and add to completeness of the data to examine the previous cold expression in soybean (which likely contains SPS gene expression) illustrated by the following:

Yamasaki et al. (RNASEQ) See GEO Accession GSE117686 

Maruyama et al, (microarray) See MIAMExpress (accession number  E-MEXP-3164). 

Why was ICE1 transcript accumulation examined?  The activity of this gene in other systems examined to date is largely regulated post-translationally. A rationale and explanation is needed somewhere in this manuscript. Also ICE1 is thought largely to be acting as a derepressor of myc (competing for its binding site) and allowing CBF to bind promoters and thus activating transcription. Are the authors suggesting a possible activation of the SPS genes by ICE1, independent of myc displacement and CBF?  If so, is there any prior support for this in any other system? Discuss extreme transience (peak in 12 h and then rapid decrease) of expression of SPS and how that may contribute to proposed changes in metabolism?? Is the ICE1 increase substantiated in other work (i.e., RNASEQ work mentioned above)? How is this increase related to much greater increase in SPS transcripts expected due to CBF regulation (cis elements are also in the promoters)? In line 364-66 it appears that the authors are suggesting an activation independent of the mechanism mentioned above. Rather than calling it a “direct” activation” a more through discussion of this is needed. 

Other details

Line 35 low temperature affects conformation of nucleic acids?? (perhaps a citation for  impact of cold on conformations of protein and nucleic acids would be appropriate here?)

Line 54-56.  This could be written more clearly

Line 87, change “K-deficient” to “K-deficiency”

Line 96: I disagree that evidence shown in this manuscript indicates DIRECT up regulation by ICE1. One does not know what other factors present on tobacco that might mediate this response. 

Line 103. Then 82 and 18 ….respectively.   please make this more clear. With respect to what>?

The following comments reflect that much of this analysis are predictions not documented or experimentally-evidence based information. Please check entire document for these types of errors. 

Line 112. Change “information” to “predictions”

Line 113, insert “predicted” between GmSPS and proteins.

Line 115 change “existed” to “were predicted”

Line 117. change “revealed” to “suggested”

Line 200. Insert “predicted” between The and phosphorylation 

Line 203 insert predicted between most and serine

Line 231. Change “Various” to “Predicted”

Line 216-218, this is all speculation, please change “may” to “could possibly” or “potentially”.

Line 126-129. Sentence needs improvement for clarity

Line 159-163, perhaps a bit more explanation or at least a direct citation that might help the reader understand the analysis; ks, Ka etc.

Figure 2 legend. Line 166. Change “These” to “The”

            It is not clear what the gray lines show.  They are not distinct “lines” so no connections can be deduced.  Need better resolution for this feature of the figure to be useful. 

Line 171, Change “showed” to “shown”.  

Line 231. Insert “which could respond” after “in”.

Line 235. insert “were also predicted” after “genes”

Line 236. Change “it is” to “they are”

Line 245. Change “leave” to “leaf”

Line 250. I disagree, all seem similar 18,4,6,13,17. Only 8 was significantly higher (based on figure 6a). I would change to “GmSPS8 had the strongest expression level in the flower”

Line 280. Here would be a good place to discuss previous omic data related to cold in soybean that either supports or not the present data (ie., Yamasaki et al. (RNASEQ) See GEO Accession GSE117686 

Maruyama et al, (microarray) See MIAMExpress (accession number  E-MEXP-3164). 

This is also relevant to both the SPS and the ICE1 expression data. 

Line 282. Change “express” to “accumulate”

Line 283. Delete “expression” and insert “transcripts” after GmSPS13/17

Line 305. This should be more weakly stated, only band shift or protection experiments coupled with mutagenesis can show direct interactions between TFs and DNA.

Line 319.  As stated above; Appropriate references should be cited rather than saying there are few research conducted on cold stress!!

Line 362, “generous” is the wrong word here. Change to “many” ?

Methods, Line 424. A better description of accession TL-1 is needed.  Where can this line be obtained? etc.

Line 442. Does GmACT11 change with cold, if so how much?

Generally Ok.  I made a few suggestions for improvement. 

Reviewer 3 Report

In the present manuscript: "Genome-wide identification of GmSPS gene family in soybean and expression analysis in response to cold stress", the authors conducted an interesting study to explore the mechanisms of the plant's to response to cold stress.

The experimental design applied it is appropriate and the results obtained are sufficiently clear. Thus, the manuscript is suitable in its present form.

Author Response

Thank you very much!

Best wishes!

Round 2

Reviewer 1 Report

The changes introduced by the authors are satisfactory, particullarly the EMSA analysis that show in vitro ICE1 binding activity. Authors should explicit that this interaction occurs in vitro, which is what demosntrates EMSA. Just mentioning it in Lines 327, 340 and 403. For in vivo demonstration ChIP analysis would be required. 

Reviewer 2 Report

This manuscript provides unique and informative data of responses of soybean to cold temperatures, particularly regarding the regulation of SPS genes.  The higher resolution figure and the EMSA experiment enhance the manuscript and provide strong support for the conclusions.

I only have a couple further suggestions:

1)    Can the authors better address Q29 regarding the line TL-1?  What is source or what line is it derived from?  Is it the same TL-1 that can be found elsewhere in the literature (which is Tianlong No.1 (TL-1, representative of shade-tolerant plants)?

2)    Line 326, change “does” to “dose”

3)    Could the authors make a statement in the manuscript (e.g., in methods) about Q30? For example “GmACT11 (Glyma.18G290800) did not change significantly in response to cold; so was used as the internal reference gene to adjust the expression level of GmSPS genes”.

Just one suggestion above.
